# Fragile X Mental Retardation Protein and Cerebral Expression of Metabotropic Glutamate Receptor Subtype 5 in Men with Fragile X Syndrome: A Pilot Study

**DOI:** 10.3390/brainsci12030314

**Published:** 2022-02-26

**Authors:** James Robert Brašić, Jack Alexander Goodman, Ayon Nandi, David S. Russell, Danna Jennings, Olivier Barret, Samuel D. Martin, Keith Slifer, Thomas Sedlak, Anil Kumar Mathur, John P. Seibyl, Elizabeth M. Berry-Kravis, Dean F. Wong, Dejan B. Budimirovic

**Affiliations:** 1Section of High Resolution Brain Positron Emission Tomography Imaging, Division of Nuclear Medicine and Molecular Imaging, The Russell H. Morgan Department of Radiology and Radiological Science, The Johns Hopkins University School of Medicine, Baltimore, MD 21287, USA; anandi1@jh.edu (A.N.); smart149@jhu.edu (S.D.M.); tsedlak@jhmi.edu (T.S.); amathur4@jh.edu (A.K.M.); dfwong@wustl.edu (D.F.W.); 2Frank H. Netter MD School of Medicine, Quinnipiac University, North Haven, CT 06473, USA; jack.goodman@quinnipiac.edu; 3Institute for Neurodegenerative Disorders, New Haven, CT 06510, USA; drussell@invicro.com (D.S.R.); jennings@dnli.com (D.J.); olivier.barret@cea.fr (O.B.); jseibyl@invicro.com (J.P.S.); 4Invicro, New Haven, CT 06510, USA; 5Denali Therapeutics, Inc., South San Francisco, CA 94080, USA; 6Laboratoire des Maladies Neurodégénératives, Molecular Imaging Research Center (MIRCen), Institut de Biologie François Jacob, Centre National de la Recherche Scientifique (CNRS), Commissariat à l’Énergie Atomique et aux Énergies Alternatives (CEA), Université Paris-Saclay, CEDEX, 92265 Fontenay-aux-Roses, France; 7Department of Neuroscience, Zanvyl Krieger School of Arts and Sciences, The Johns Hopkins University, Baltimore, MD 21218, USA; 8Department of Psychiatry and Behavioral Sciences-Child Psychiatry, The Johns Hopkins University School of Medicine, Baltimore, MD 21205, USA; slifer@kennedykrieger.org; 9Department of Behavioral Psychology, Kennedy Krieger Institute, Baltimore, MD 21205, USA; 10Department of Psychiatry and Behavioral Sciences-General Psychiatry, The Johns Hopkins University School of Medicine, Baltimore, MD 21205, USA; 11Departments of Pediatrics, Neurological Sciences, and Biochemistry, Rush University Medical Center, Chicago, IL 60612, USA; elizabeth_berry-kravis@rush.edu; 12Laboratory of Central Nervous System (CNS) Neuropsychopharmacology and Multimodal, Imaging (CNAMI), Mallinckrodt Institute of Radiology, Washington University, Saint Louis, MO 63110, USA; 13Department of Psychiatry, Kennedy Krieger Institute, Baltimore, MD 21205, USA

**Keywords:** anterior cingulate cortex, correlation coefficient, fragile X mental retardation 1 gene (*FMR1*), linear regression, neurodevelopmental disorders, neuroimaging, temporal cortex, positron emission tomography (PET), radiotracer, thalamus

## Abstract

Multiple lines of evidence suggest that a deficiency of Fragile X Mental Retardation Protein (FMRP) mediates dysfunction of the metabotropic glutamate receptor subtype 5 (mGluR_5_) in the pathogenesis of fragile X syndrome (FXS), the most commonly known single-gene cause of inherited intellectual disability (ID) and autism spectrum disorder (ASD). Nevertheless, animal and human studies regarding the link between FMRP and mGluR_5_ expression provide inconsistent or conflicting findings about the nature of those relationships. Since multiple clinical trials of glutamatergic agents in humans with FXS did not demonstrate the amelioration of the behavioral phenotype observed in animal models of FXS, we sought measure if mGluR_5_ expression is increased in men with FXS to form the basis for improved clinical trials. Unexpectedly marked reductions in mGluR_5_ expression were observed in cortical and subcortical regions in men with FXS. Reduced mGluR_5_ expression throughout the living brains of men with FXS provides a clue to examine FMRP and mGluR_5_ expression in FXS. In order to develop the findings of our previous study and to strengthen the objective tools for future clinical trials of glutamatergic agents in FXS, we sought to assess the possible value of measuring both FMRP levels and mGluR_5_ expression in men with FXS. We aimed to show the value of measurement of FMRP levels and mGluR_5_ expression for the diagnosis and treatment of individuals with FXS and related conditions. We administered 3-[^18^F]fluoro-5-(2-pyridinylethynyl)benzonitrile ([^18^F]FPEB), a specific mGluR_5_ radioligand for quantitative measurements of the density and the distribution of mGluR_5_s, to six men with the full mutation (FM) of FXS and to one man with allele size mosaicism for FXS (FXS-M). Utilizing the seven cortical and subcortical regions affected in neurodegenerative disorders as indicator variables, adjusted linear regression of mGluR_5_ expression and FMRP showed that mGluR_5_ expression was significantly reduced in the occipital cortex and the thalamus relative to baseline (anterior cingulate cortex) if FMRP levels are held constant (F(7,47) = 6.84, *p* < 0.001).These findings indicate the usefulness of cerebral mGluR_5_ expression measured by PET with [^18^F]FPEB and FMRP values in men with FXS and related conditions for assessments in community facilities within a hundred-mile radius of a production center with a cyclotron. These initial results of this pilot study advance our previous study regarding the measurement of mGluR_5_ expression by combining both FMRP levels and mGluR_5_ expression as tools for meaningful clinical trials of glutamatergic agents for men with FXS. We confirm the feasibility of this protocol as a valuable tool to measure FMRP levels and mGluR_5_ expression in clinical trials of individuals with FXS and related conditions and to provide the foundations to apply precision medicine to tailor treatment plans to the specific needs of individuals with FXS and related conditions.

## 1. Introduction

Fragile X syndrome (FXS), the leading monogenetic cause of intellectual disability (ID) and autism spectrum disorder (ASD), results from excessive trinucleotide cytosine–guanine–guanine (CGG_n_) repeats in the promotor region [1,2,3,4] of the fragile X mental retardation 1 (*FMR1*) gene. The *FMR1* gene leads to the development in healthy humans with typical development (TD) of Fragile X Mental Retardation Protein (FMRP) [1,5,6,7], an RNA binding protein playing key roles in the protein synthesis of dendritic spines by controlling 4% of human cerebral mRNA translation [1,8,9,10]. The normal sequence of the *FMR1* gene facilitating the growth of FMRP throughout the body is accomplished in healthy individuals with TD by means of a normal number of trinucleotide CGG_n_ repeats in the promotor region [1,2,3,4] of the *FMR1* gene. Healthy individuals with typical development (TD) have a CGG repeat sequence of approximately 6 to 54 [11] categorized as low zone (<24 CGG), normal (24–42 CGG), and gray zone (42–54 CGG) [12]. Individuals with 55 to 200 CGG repeats are given premutation (PM) or carrier status [13,14,15,16] leading to fragile X tremor and ataxia syndrome (FXTAS) or fragile X primary ovarian insufficiency (FXPOI) [12]. An intermediate zone of approximately 45 to 60 repeats characterizes individuals between the typical and PM status [17]. The full mutation (FM), clinical FXS, is characterized by more than 200 CGG repeats, resulting in epigenetic silencing of the *FMR1* gene via hypermethylation leading to a marked deficiency of the gene’s product: FMRP. Studies with *Fmr1* knockout (KO) rodent models demonstrate that FMRP is required for neuronal dynamics at multiple levels in the auditory [18], limbic [19], and sensory systems [20]. Furthermore, FMRP suppresses translation of the endoplasmic reticulum stress response augmented by amyloid beta (Aβ), a toxic peptide that accumulates in the brains of people with Alzheimer’s disease and other cognitive deficits [21].

Glutamate, another agent playing a role in stress responses throughout the body, is the major excitatory neurotransmitter. Glutamate is needed for the development and function of neurons throughout the nervous system. The vesicular glutamate transporter takes glutamate in presynaptic neurons to vesicles to be released into the synapse and to act on receptors in post-synaptic neurons to generate signals to activate the neurons [22]. Glutamine acts upon both (A) ionotropic receptors including *N*-methyl-d-aspartate (NMDA, α-amino-3-hydroxy-5-methyl-4-isoxazolepropionic acid (AMPA), and kainic acid (KA) receptors, and (B) metabotropic glutamate receptors. Additionally, kyurenic acid, a noncompetitive antagonist of the *N*-methyl-d-aspartate (NMDA) glutamate receptor, plays a role in the release of glutamate, dopamine, and acetylcholine [22,23]. Dysfunction of glutamatergic neurotransmission plays a role in mood disorders [22,23], Alzheimer’s disease [24,25] and other neuropsychiatric disorders. The development of pharmacological agents for the glutamatergic systems that offer promise for multiple neuropsychiatric disorders [26] provide the motivation for continued efforts to develop glutamatergic agents to alleviate the symptoms and signs of FXS.

To utilize the key role of the glutamatergic system in the understanding of the pathogenesis of FXS, the mGluR theory was postulated on the observation that activation of group I metabotropic glutamate receptors (metabotropic glutamate receptors subtypes 1 and 5 (mGluR_1/5_)) [27] is associated with dysregulated downstream signaling cascades [28] based on the loss of the regulation of translation in the absence of FMRP. The affected signaling cascades include the mammalian target of rapamycin (mTOR) [29,30,31,32,33], the microtubule-associated protein kinase (MAPK) [34], the protein kinase A (PKA) [35], and the extracellular signal-regulated kinase (ERK) pathways, which may contribute to metabotropic glutamate receptor dependent long-term depression (mGluR-LTD) and to the neurobehavioral symptoms of FXS in *Fmr1* knockout (KO) mouse models [7,15,16,36,37,38]. Although the mGluR theory hypothesizes a possible mechanism for FXS in *Fmr1* KO mouse models [30,39], mGluR-based clinical trials have not yielded improvements in the behavioral phenotype of FXS in humans [39,40,41]. The addition of measurements of FMRP levels to the measurements of mGluR expression in the relevant regions of the brains of men with FXS may facilitate future clinical trials for FXS. The linkage of FMRP values and the expression of metabotropic glutamate receptors subtype 5 (mGluR_5_) in the living human brain could provide a tool to ameliorate cognitive and behavioral symptoms associated with FXS such as ASD [2,6,7,41,42].

Nevertheless, the role of metabotropic glutamate receptors subtype 5 (mGluR_5_) in the expression of the neurobehavioral phenotype of FXS [7,8,15,16] is unknown. Although human autopsy and animal imaging studies have reported inconsistent values for mGluR_5_ [41,42], mGluR_5_ expression was decreased in studies of KO mouse models of FXS [43]. Additionally, reduced protein synthesis in cerebral [44] and peripheral measurements [45] in FXS raises questions to the mGluR theory [9] that require further clarification.

Although the behavioral benefits of negative allosteric modulators (NAMs) in animal models of FXS [46] were not translated in multiple clinical trials of humans with FXS [40,41], there may be different mechanisms in animal models of FXS and humans with FXS [47]. Since NAMs bind allosterically to mGluR_5_s as noncompetitive antagonists, reduced post-synaptic excitation [48] may decrease mGluR_5_ expression in humans with FXS. Furthermore, decreased connectivity between mGluR_5_s and Homer proteins, the primary members of the post-synaptic density (PSD) connecting mGluR_5_s to their signaling complexes [49], may occur in FXS. The positive behavioral effects of deletion of the gene responsible for the irregular mGluR_5_–Homer scaffold interactions supports a relationship to the phenotype in FXS [50]. Tolerance following chronic treatment with NAMs, acquired treatment resistance downstream of glycogen synthetase kinase 3α (GSK3α) and upstream of protein synthesis [51,52,53], may play a role in the negligible response to NAMs in human clinical trials. Chronic treatment of *Fmr1* KO mice with an inhibitor of GSK3α and GSK3β reversed the social discrimination deficit observed in the mice [54]. While some recent data in humans with FXS suggested decreased protein synthesis was observed in cerebral [44] and peripheral measurements [45], the majority of prior studies have found increased protein synthesis in the cells of people with FXS [5,55].

Clinical trials of mGluR_5_ NAMs for FXS were limited by the variable age groups of study participants, problematic outcome measurements, placebo effects, and potentially the absence of a tool to measure the expression of mGluR_5_ in the living brains of participants with FXS [2,42,43,44,46,56,57]. Our finding that mGluR_5_ expression measured by positron emission tomography (PET) with 3-[^18^F]fluoro-5-(2-pyridinylethynyl)benzonitrile ([^18^F]FPEB), a potent and safe specific mGluR_5_ inhibitor (Figure 1 and Figure 2) [58,59], is reduced in all brain regions in men with FXS [2] compared to participants of both sexes with ASD and TD [2,42,60,61,62], was confirmed utilizing positron emission tomography and magnetic resonance (PET/MR) imaging on an unmedicated cohort of older men with FXS and age- and sex-matched participants with TD suggesting that the current protocol may be a valuable biomarker for mGluR_5_ expression in clinical trials of novel agents for humans with FXS and other subtypes of ASD [63]. We showed that [^18^F]FPEB may be a promising tool to obtain quantitative measurements of mGluR_5_ expression in individuals with FXS for clinical trials and other investigations [2,42,64,65,66,67]. Since our prior finding of reduced cerebral mGluR_5_ expression in cortical and subcortical regions of men with FXS has been confirmed [41,63], we sought expand the previous protocol of cerebral mGluR_5_ expression alone with a more refined measure tool to include simultaneous FRMP levels and cerebral mGluR_5_ expression for clinical trials of FXS. Here, we seek to investigate the relationship between FMRP [68] and mGluR_5_ expression in unmedicated men with the FM of FXS [15,16].

Development of interventions to ameliorate the specific molecular deficits of individuals with FXS may facilitate the development of treatment plans to address the unique needs of each person [41,42,64,71,72,73]. We aimed to determine if (A) FMRP levels are correlated with mGluR_5_ expression in men with the FM of FXS [2,41,42] and (B) the measurement of FMRP and PET with ([^18^F]FPEB) is feasible in men with FXS [2,41,42,60,61]. The proposed protocol could then be utilized as a screening measure in pilot studies to optimize dosing for people with FXS and related conditions.

## 2. Materials and Methods

### 2.1. Participants

#### 2.1.1. Recruiting Sites

The study is approved by Johns Hopkins Medicine Institutional Review Board IRB 169,249 [2]. The protocols for the study of humans were approved by the Institutional Review Boards of the Institute for Neurodegenerative Disorders (IND) in New Haven, Connecticut [74] and the Johns Hopkins University (JHU) in Baltimore, Maryland [75,76]. Since exposure to radioactivity in PET constitutes greater than minimal risk, this pilot study was restricted to adults [2,42]. Written informed consent was obtained from each participant at both locations.

Although we have reported the data of eight participants from IND and four participants from JHU [2,42,60,64,65,66,67,71,72], we currently report only the finding of six men with the FM of FXS and one man with FXS-M with a premutation allele of 181–189 CGG repeats close to the allele size of people with the FM (>200 CGG repeats) from IND. We excluded the data of men from JHU due to the uncertain effects of the markedly different protocols at IND and JHU. We excluded another participant from IND due to missing data. We now report the findings of a cohort at the IND of six Caucasian males with the FM of FXS (mean age 27.8 ± 4.7, range (22.3, 33.6) years) and a Caucasian man with an FXS-M (age 56.6 years) [2,42,60,62,64,65,66,67,71,72,73] (Table 1) [62].

#### 2.1.2. Inclusion Criteria

Inclusion criteria for all participants included age between 18 and 60 years [2,42,74]. Participants with FXS had a diagnosis of the FM or FXS-M of FXS based on *FMR1* DNA gene testing by polymerase chain reaction (PCR)/Southern Blot [77] on peripheral venous blood samples [11,15,16], supplemented by clinical neurobehavioral profiling [2,7,42,62].

#### 2.1.3. Exclusion Criteria

Exclusion criteria were clinically significant abnormal laboratory values and/or clinically significant unstable serious medical, neurological, or psychiatric illnesses [2,42,60,62,74].

### 2.2. Procedures

#### 2.2.1. Fragile X Mental Retardation Protein (FMRP)

##### Enzyme-Linked Immunoassay (ELISA)

Primary lymphocytes or fibroblasts were quickly thawed and spun at 2000× *g* for 10 min. Pelleted cells were resuspended in phosphate-buffered saline (PBS) containing protease inhibitor tablet and washed two times more. Cells were lysed in the presence of protease inhibitors, rotated overnight at 4 °C and spun at 16,000× *g* for 15 min. Supernatant was removed and aliquoted for storage at −80 °C. Quantitation of total protein concentration was accomplished with a bicinchoninic acid protein assay kit. Then, 96-well plates were coated with a chicken antibody generated to the peptide sequence near the carboxy terminus, KDRNQKKEKPDSVD (Aves Labs, Inc. Tigard, OR, USA), and 100 uL of lymphocyte extract or FMRP was added to the prepared wells and incubated overnight at room temperature. Following extensive washing, detection antibody, 1:10,000 (v:v) mouse anti-FMRP (1C3, MilliporeSigma, Temecula, CA, USA), was added to each well and incubated for 8–10 h at room temperature. Wells were washed and incubated with peroxidase conjugated secondary antibody to mouse IgG. Detection was accomplished with a luminescent peroxidase substrate. All lymphocyte samples were assessed at multiple dilutions, and concentrations of FMRP were calculated relative to a reference sample of purified FMRP of known concentration using a 4-parameter fit logistics curve. Coefficients of variation were calculated to assess reliability of ELISA measurements [78].

##### Assay by Luminex Technology

Primary lymphocytes were quickly thawed and spun at 4000× *g* for 10 min. Pelleted cells were resuspended in phosphate-buffered saline (PBS) containing protease inhibitor tablet (Roche, Basel, Switzerland) and washed two times more. Cells were lysed in the presence of protease inhibitors, antipain, and chymostatin, rotated overnight at 4 °C and spun at 16,000× *g* for 20 min. Supernatant was removed and aliquoted for storage at −80 °C. Quantitation of total protein concentration was accomplished with a bicinchoninic acid protein assay kit. Then, 4 ug of cell lysate was suspended in Luminex buffer (PBS, 1% BSA, 0.05% Tween 20, 0.05% NaAzide) and approximately 3000 serological beads were coated with capture antibody. Mouse monoclonal 6B8 were loaded in 96-well filter plates and incubated on vortex shaker for 6 h. Rabbit anti-FMRP R477 antibody was added 1:625 (v:v) to each well and incubated overnight in 4 °C. Goat anti-rabbit IgG conjugated to phycoerythrin antibody (Thermo Scientific, Waltham, MA, USA) was added 1:250 (v:v) and incubated for 2 h. Samples were read on a Luminex 200 machine counting 50 events/bead region (region 33). All lymphocyte samples were assessed with duplicates and the concentration of FMRP was calculated relative to a standard reference sample of a recombinant fusion protein carrying short domains of FMRP, GST-SR7 at known concentration. Coefficients of variation were calculated to assess reliability of Luminex measurements [79]. 

#### 2.2.2. Positron Emission Tomography (PET)

Participants received training by behavioral psychologists including mock scanner training before the procedure. All participants underwent scans conducted by an experienced research staff of Certified Nuclear Medicine Technologists (CNMT) who had attained certification by the Nuclear Medicine Technology Certification Board (NMTCB). Before conducting these scans, the technologists had conducted many PET scans with participants who experienced challenges to maintain stillness throughout scans. The technologists maintained the physical conditions of each scan optimally for the completion of the scans. Participants were positioned by the technologists in the most comfortable manner for scans. Heads were stabilized in the scanner by gauge strips [2,42,74]. In order to maintain a comfortable environment during the scans, technologists utilized blankets and pads to raise legs. The physical conditions of the scans were maintained in optimal manners for participants by outstanding technologists.

Positron emission tomography (PET) after the intravenous bolus injection 185 MBq (5 mCi) of [^18^F]FPEB [2,42,60,61,62,63,64,65,66,67,80] was conducted on an ECAT EXACT HR+ PET manufactured by Siemens/CTI (Knoxville, TN, USA) [81] for 90–120 min after injection. Injectors obtained measured doses of [^18^F]FPEB synthesized by radiochemists in the adjacent radiochemistry laboratory following the published methods [60] to be administered to participants in the scanning chambers.

#### 2.2.3. Statistical Analyses

Statistical Parametric Mapping (SPM) [70] was applied to PET frames to obtain regional time activity (radioactivity) curves (TACs) in regions often affected in neurodegenerative disorders. Standard uptake values (SUVs) of ratios of radiotracer uptake in the region of interest to the cerebellum, a region with minimal radio tracer uptake [82] were generated (Table 2).

The Pearson correlation coefficient between the FMRP and the [^18^F]FPEB uptake in the ROIs was calculated with a significance level of 0.05 in six men with the FM and one man with the FXS-M of FXS [83].

Linear regression of [^18^F]FPEB uptake on FMRP was examined in men with the FM and FXS-M of FXS (StataCorp. 2021. Stata Statistical Software: Release 17. College Station, TX, USA: StataCorp LLC.).

## 3. Results

None of the regions demonstrated statistical significance for the Pearson correlation coefficient between the FMRP and the [^18^F]FPEB uptake in the ROIs of the six men with the FM of FXS (Table 3) [83].

The addition of data from the man with an FXS-M yielded Pearson correlation coefficients that did not attain statistical significance (Table 4). However, the findings suggest that a larger sample may attain significance.

In order to clarify the roles of individual brain regions on the association of FMRP and [^18^F]FPEB uptake, we performed a simple linear regression for brain regions leading to a significant result [*p* < 0.001] (StataCorp. 2021. Stata Statistical Software: Release 17. College Station, TX, USA: StataCorp LLC.) (Table 5). Simple linear regression demonstrated a significant association [F(1,47) = 9.87, *p* < 0.001] (Table 5). Utilizing the seven cortical and subcortical regions affected in neurodegenerative disorders as indicator variables, adjusted linear regression of mGluR_5_ expression and FMRP (Table 6) showed that mGluR_5_ expression was significantly reduced in the occipital cortex and the thalamus relative to baseline (Anterior cingulate cortex [ACC]) if FMRP levels are held constant [F(7,47) = 6.84, *p* < 0.001] (Table 5) (Figure 3).

## 4. Discussion

In order to generate a secure foundation to guage the effects of glutamatergic agents in clinical trials of men with FXS, we have demonstrated the value of simultaneous measurements of FMRP levels and mGluR_5_ expression in relevant brain regions. We have shown the benefit to include FMRP levels along with our previously reported protocol for mGluR_5_ expression in relevant brain regions of men with FXS. The current study expands and enhances the extent of our prior report using mGluR_5_ expression in relevant brain regions of men with FXS for clinical trials (2,42). Among a small sample of men with the FM and FXS-M of FXS there is a significant association between the FMRP levels and the mGluR_5_ expression. Additionally, the anterior cingulate cortex [2], the occipital cortex, and the thalamus modify this relationship by decreasing mGluR_5_ expression (Figure 3; Table 5). We anticipate that future studies with adequate sample sizes will have the power to confirm a positive association between FMRP levels and mGluR_5_ expression in more brain regions in men with the FM of FXS. Due to the negligible levels of FMRP in men with the FM, we included in our analyses data from a man with an FXS-M with a premutation allele of 181–189 CGG repeats close to the allele size of people with the FM (>200 CGG repeats) because he contributed a larger value of FMRP. We sought to avoid the confounding effects of the autopsy specimens of individuals with FMs and premutations (PMs) [84,85]. The current study advances the foundations for clinical trials of glutamatergic agents for FXS by adding the objective quantification of FMRP levels to the measurement of mGluR_5_ expression in relevant brain regions. We provide the tools for sound protocols of glutamatergic agents for FXS.

These findings suggest that mGluR_5_ expression may be related to FMRP levels to play a role in the pathogenesis of FXS. The protocol for this investigation provides a feasibility tool that may facilitate the measurement of a biomarker of mGluR_5_ expression to conduct rigorously designed clinical trials of FXS [2,40,41] and perhaps other subtypes of ASD [42]. That said, the findings of this study merit replication in a larger sample of the groups studied here and other neurodevelopmental disorders [86]. The proposed protocol may provide a tool to assess the role of FMRP and mGluR_5_ expression in other subtypes of ASD [87], schizophrenia [88], and mood disorders [89]. Indeed, the current protocol may be expanded to promote knowledge about multiple neuromodulators in FXS, Rett syndrome [90,91], and other subtypes of ASD [92].

The proposed protocol is particularly important as a means to assess individuals with FXS and related conditions in community settings within a hundred miles of a production center with a cyclotron. Samples of blood can be drawn to be sent on dry ice for FMRP analysis at specialized research laboratories. Since [^18^F]FPEB is an [^18^F] compound with a half-life of two hours, it can be manufactured in specialized facilities for distribution two hours away. Thus, [^18^F]FPEB, like 2-deoxy-2 [^18^F]fluoro-D-glucose ([^18^F]FDG), the most widely used radiotracer, can be utilized for scans in community settings with portable PET scanners and nuclear medical scanners modified to conduct PET scans in proximity to production centers with cyclotrons. Unlike [^11^C] radiotracers with a half-life of around 20 min that require manufacture in a cyclotron adjacent to the scanner, [^18^F]FPEB, like [^18^F]FDG, can be commercially manufactured by central facilities with cyclotrons for distribution to community settings within a radius of a hundred miles [93]. Thus, the proposed protocol has the potential for commercial development to obtain data for sophisticated analyses by experts in tertiary centers.

The current status of glutamatergic interventions for FXS resembles the status of amyloid agents for Alzheimer’s disease years ago. Now three [^18^F] radiotracers for amyloid are commercially available for use before the administration of novel agents to reduce amyloid. The proposed protocol could be utilized for clinical trials of glutamatergic agents for FXS.

The current investigation is limited by the utilization of data from brain regions conjectured to be affected in men with the FM or an FXS-M of FXS utilizing SUV analysis of PET scans only with an established analytic technique for MR scan [70]. Since regions were selected for this study based on our experience with neurodegenerative disorders, key anatomical regions likely to be affected in FXS were not included. The neurobehavioral phenotype of FXS is associated with delayed socialization reflecting cognitive processes in cortical regions and avoidance reflecting limbic circuits [94]. Utilization of the same protocol to obtain MR and PET scans on all participants will enhance the accuracy of future investigations. Coregistration of PET and MR images will facilitate the analysis of future investigations. The regions chosen for this analysis were those commonly affected in neurodegenerative disorders. Future investigations will be enhanced by precise description of the delineation of each region (e.g., the entire caudate nucleus or just the head of the caudate nucleus, which part of each cortical region or the entire cortex). Comparing and contrasting of mGluR_5_ expression in participants with FXS, related conditions, and healthy participants with TD, particularly in insular, temporal, and cingulate cortices, the regions with high of mGluR_5_ expression in healthy men with TD [60], will greatly enhance anatomical localization of mGluR_5_ expression in FXS and related conditions. Future investigations to identify characteristic structures related to (A) higher cortical functions will include the prefrontal cortex and (B) learning and emotions will include the hippocampus and the amygdala. Future work will include measurement of inhibitory control [95], mediofrontal negativity [96], and other additional variables likely to be impaired in men with FXS.

Future investigations will be enhanced by contemporaneous conduct of all investigations at all participating institutions with identical protocols and analyses [2,42]. Since full maturation of the human brain may continue through the ages of 25 and 30, it is possible that expression of mGluR_5_ in these participants is still influenced by developmental processes. Therefore, future studies of older individuals may better identify the characteristics of mGluR_5_ expression in the brains of mature individuals. Additionally, the age differences may represent a confounding influence because increased age is associated with decreased cerebral expression of mGluR_5_ measured by PET with [^18^F]FPEB [97]. The variability of BMIs may represent another confounding influence [2,42,62].

Diurnal variation may have influenced mGluR_5_ expression in the different scans utilized for this study. The ranges of injection times were (1100, 1543). Since changes in mGluR_5_ expression vary as much as 30% in two hours [2,98], diurnal variation may have influenced the variability of mGluR_5_ expression in our cohort. Performing scans at exactly the same time in all participants is desirable. Another limitation is our use of ELISA in lymphocytes to estimate FMRP expression as a function of brain region. Peripheral FMRP expression may differ from FMRP expression in various brain regions. Utilization of only participants with the full mutation of FXS will eliminate the errors introduced with participants with PMs and other genetic variations [6,84].

Multicenter PET/MR studies on unmedicated individuals with FXS [99] with optimal qualification [100] provide a tool to obtain optimal target engagement measurements [67] without the unknown influences of concomitant medications [101]. In order to avoid the marked diurnal variations in mGluR_5_ expression [2,98], scans may be performed at exactly the same time in all participants. Since dysfunction of the amygdala and the rectus gyrus underlie anxiety in individuals with FXS [102,103], especially in females with FXS [104], future investigations of these regions in females and males with FXS and other subtypes of ASD may provide foundations for interventions to reduce anxiety in affected individuals. The proposed procedure may be developed to facilitate the diagnosis and treatment of individuals with FXS and other subtypes of ASD early in life [105,106,107]. Prompt procedures to rule out competing items of the differential diagnosis leading to confirmation of the diagnosis of FXS and other subtypes of ASD will facilitate the basis to institute optimal educational placement and other needed interventions [108]. This protocol will provide the foundations of trustworthy evidence [109] to disseminate widely with the general public [110].

The current investigation demonstrates the usefulness of the proposed protocol to measure FMRP levels and of mGluR_5_ expression in brain regions relevant for FXS to provide sound foundations for quantitation before, during, and after clinical trials of glutamatergic agents. We have demonstrated that the proposed protocol including challenging physical and psychological tasks in is feasible in individuals with FXS and other disabilities by means of training by behavioral psychologists before and during the tasks. The ultimate goal of this work is to utilize the proposed protocol for large-scale multicenter studies of potential interventions for FXS. We seek to expand this protocol to develop optimal means for the diagnosis, treatment, cure, and prevention of FXS and related conditions.

## 5. Conclusions

We showed an association of FMRP levels and cerebral mGluR_5_ expression measured by PET with [^18^F]FPEB in the anterior cingulate cortex, the occipital cortex, and the thalamus in men with the FM or FXS-M of FXS. Reduced cortical mGluR_5_ expression in these cortical regions may provide a basis for the severity of the neurobehavioral phenotype of cognitive deficits and delayed socialization of individuals with FXS [108]. Reduced limbic mGluR_5_ expression may provide a basis for the avoidance behaviors of individuals with FXS [7,94]. Since the effects of interventions on animal models of FXS may not produce similar effects in humans with FXS [111], current theories of FXS merit reexamination [44]. The proposed protocol may provide a crucial tool to measure mGluR_5_ expression in clinical trials of peptides [26] and other novel interventions in FXS. Our findings may provide the foundations to generate novel theories of humans with FXS.

The proposed protocol may provide a biomarker for measurement of FMRP and mGluR_5_ expression of relevance for clinical trials of FXS and other subtypes of ASD. The proposed protocol may provide a tool to foster diagnostic and therapeutic interventions for FXS and related conditions.

## Figures and Tables

**Figure 1 brainsci-12-00314-f001:**
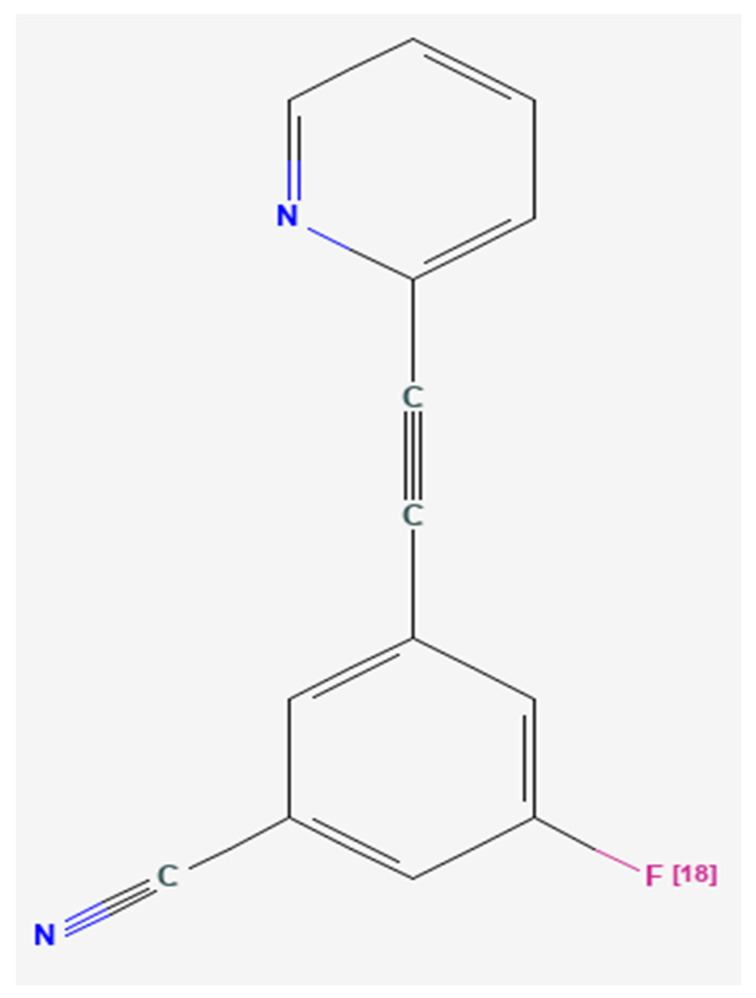
Chemical structure of 3-[^18^F]fluoro-5-(2-pyridinylethynyl)benzonitrile ([^18^F]FPEB) [58], a potent, specific inhibitor of the metabotropic glutamate receptor subtype 5 (mGluR_5_) [60].

**Figure 2 brainsci-12-00314-f002:**
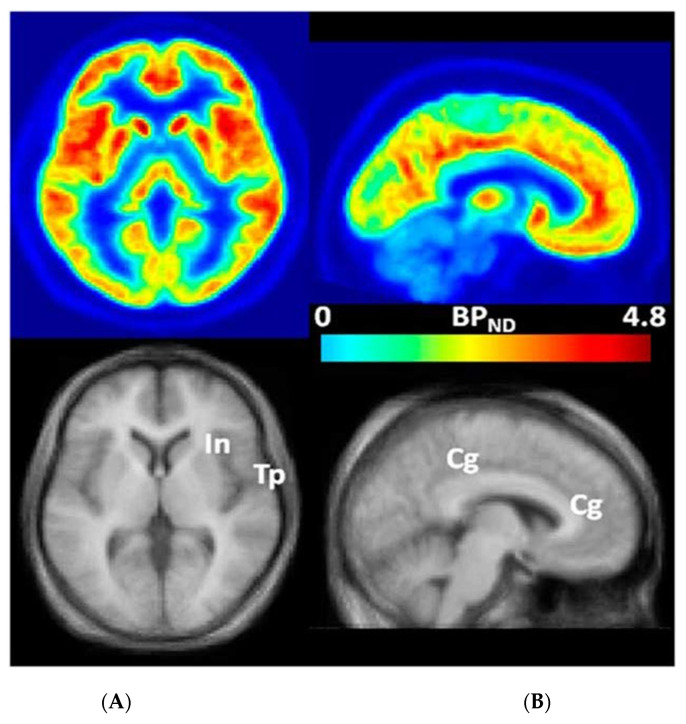
Transaxial (**A**) and sagittal (**B**) non-displaceable binding potential (BP_ND_) [69] images of 3-[^18^F]fluoro-5-(2-pyridinylethynyl)benzonitrile ([^18^F]FPEB) (top) and matching magnetic resonance (MR) images (bottom) in statistical parametric mapping (SPM) [60,70] standard space. Regions with high BP_ND_ values, namely, insular (In), temporal (Tp), and cingulate (Cg) cortices, are indicated on co-registered MR images [60,70]. This research was originally published in *JNM*. Wong, D.F.; Waterhouse, R.; Kuwabara, H.; Kim, J.; Brašić, J.R.; Chamroonrat, W.; Stabins, M.; Holt, D.P.; Dannals, R.F.; Hamill, T.G.; Mozley, P.D. ^18^F-FPEB, a PET radiopharmaceutical for quantifying metabotropic glutamate 5 receptors: A first-in-human study of radiochemical safety, biokinetics, and radiation dosimetry. *J. Nucl. Med.*
**2013**, *54*, 388–396. © SNMMI [60].

**Figure 3 brainsci-12-00314-f003:**
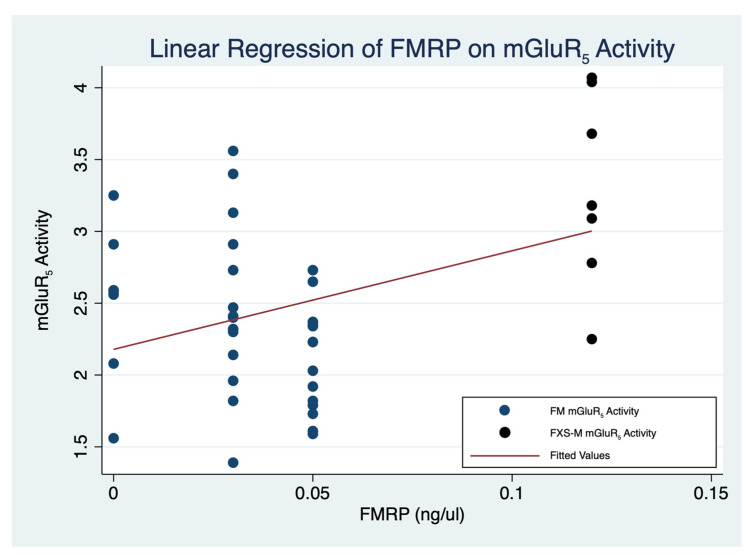
Linear regression of 3-[^18^F]fluoro-5-(2)pyridinylethynyl)benzonitrile ([^18^F]FPEB) uptake on FMRP in men with a FM and an FXS-M of FXS (F(7,47) = 6.84, *p* < 0.001) (StataCorp. 2021. Stata Statistical Software: Release 17. College Station, TX: StataCorp LLC.) (Table 5). FM: Full mutation; FMRP: Fragile X Mental Retardation Protein; FXS: Fragile X syndrome; FXS-M: FXS-M: Allele size mosaicism of fragile X syndrome.

**Table 1 brainsci-12-00314-t001:** Clinical characteristics of Caucasian male participants with the full mutation (FM) or allele size mosaicism (FXS-M) (age = 56.6) of fragile X syndrome (FXS) [62].

Age in Years	BMI	Concomitant Medications	FM Size (kb)	Percent Methylation of FM	Mutation Size (Repeats)	FMRP	Reading Level
22.3	33.2	Clonazepam Clonidine Lamotrigine Levothyroxine sodium Potassium chloride Quetiapine	2.3–3.8	100	760–1260	0.03	<first grade
24.3	30.9	Multivitamins	3.3–4.8	100	1100–1600	0.05	<first grade
26.4	24.1	Calcium Multivitamins Sertraline	3.3–4.8	100	1100–1600	0.05	<first grade
26.8	25.8	Methylphenidate	1.8–3.8	100	600–1260	0.00	<first grade
33.3	22.0	Imodium Sertraline	1.8–4.8	100	600–1600	0.03	<first grade
33.6	.	Multivitamins	2.8–3.3	90	930–1100	0.12	<first grade
56.6	34.1	Naproxen sodium	0.8	100	260	0.24	eighth grade

BMI: Basal metabolic index; FM: Full mutation; FMRP: Fragile X Mental Retardation Protein in nanogram per microgram total protein; kb: Kilobase.

**Table 2 brainsci-12-00314-t002:** Fragile X Mental Retardation Protein (FMRP) in nanogram per microgram total protein and ratios of 3-[^18^F]fluoro-5-(2-pyridinylethynyl)benzonitrile ([^18^F]FPEB) uptake in the regions of interest (ROIs) to uptake in the cerebellum of men with the full mutation (FM) or an allele size mosaicism (FXS-M) (age = 56.6) of fragile X syndrome (FXS) [62].

Age in Years	FMRP	ACC	PCC	PC	TC	OC	S	T
22.3	0.03	2.30	2.40	1.96	2.14	1.82	2.41	1.39
24.3	0.05	2.34	1.79	1.82	1.92	1.59	2.35	1.61
26.4	0.05	2.65	2.23	2.23	2.37	2.03	2.73	1.73
26.8	0.00	2.58	2.56	3.25	2.59	2.08	2.91	1.56
33.3	0.03	3.40	2.91	2.73	3.13	2.47	3.56	2.32
33.6	0.12	4.07	3.18	3.09	3.68	2.78	4.04	2.25
56.6	0.24	3.51	2.82	2.77	3.53	2.73	3.76	2.19

ACC: Anterior cingulate cortex; FMRP: Fragile X Mental Retardation Protein; OC: Occipital cortex; PC: Parietal cortex; PCC: Posterior cingulate cortex; S: Striatum; T: Thalamus; TC: Temporal cortex.

**Table 3 brainsci-12-00314-t003:** Pearson correlation coefficient R between Fragile X Mental Retardation Protein (FMRP) in nanogram per microgram total protein and ratios of 3-[^18^F]fluoro-5-(2-pyridinylethynyl)benzonitrile ([^18^F]FPEB) uptake in the regions of interest (ROIs) to uptake in the cerebellum of men with the full mutation (FM) of fragile X syndrome [83].

Region	Pearson Correlation Coefficient of FMRP and mGluR_5_ Uptake	Probability *p* Value
Anterior cingulate cortex	0.705	0.117
Posterior cingulate cortex	0.388	0.447
Parietal cortex	0.085	0.873
Temporal cortex	0.569	0.239
Occipital cortex	0.542	0.266
Striatum	0.573	0.231
Thalamus	0.535	0.274

FMRP: Fragile X Mental Retardation Protein; mGluR_5_: Metabotropic glutamate receptor subtype 5.

**Table 4 brainsci-12-00314-t004:** Pearson correlation coefficient R between Fragile X Mental Retardation Protein (FMRP) in nanogram per microgram total protein and ratios of 3-[^18^F]fluoro-5-(2-pyridinylethynyl)benzonitrile ([^18^F]FPEB) uptake in the regions of interest (ROIs) to uptake in the cerebellum of men with the full mutation (FM) or an allele size mosaicism (FXS-M) of fragile X syndrome [83].

Region	Pearson Correlation Coefficient of FMRP and mGluR_5_ Uptake	Probability *p* Value
Anterior cingulate cortex	0.605	0.150
Posterior cingulate cortex	0.392	0.384
Parietal cortex	0.194	0.677
Temporal cortex	0.660	0.107
Occipital cortex	0.657	0.109
Striatum	0.614	0.142
Thalamus	0.561	0.190

FMRP: Fragile X Mental Retardation Protein; mGluR_5_: Metabotropic glutamate receptor subtype 5.

**Table 5 brainsci-12-00314-t005:** Linear regression of 3-[^18^F]fluoro-5-(2)pyridinylethynyl)benzonitrile ([^18^F]FPEB) uptake on FMRP in men with a FM and an FXS-M of FXS with and without adjustment set to [^18^F]FPEB uptake in the anterior cingulate cortex (ACC) as the baseline with standard errors in parentheses (StataCorp. 2021. Stata Statistical Software: Release 17. College Station, TX, USA: StataCorp LLC.) (Figure 3).

Variable	Unadjusted	Probability *p* Value	Adjusted	Probability *p* Value
FMRP	3.64 (1.16)	0.001	3.64 (0.93)	0.001
Posterior cingulate cortex			−0.42 (0.26)	NS
Parietal cortex			−0.43 (0.26)	NS
Temporal cortex			−0.21 (0.26)	NS
Occipital cortex			−0.76 (0.26)	0.001
Striatum			0.13 (0.26)	NS
Thalamus			−1.11 (0.26)	0.001
Constant	2.31 (0.12)	0.001	2.71 (0.20)	0.001
Observations	49		49	
R−squared	0.17		0.54	

NS: Not significant.

**Table 6 brainsci-12-00314-t006:** Code to adjust linear regression of mGluR_5_ expression and FMRP utilizing seven cortical and subcortical regions affected in neurodegenerative disorders as indicator variables (StataCorp. 2021. Stata Statistical Software: Release 17. College Station, TX, USA: StataCorp LLC.)

rename anteriorcingulatecortex y1
. rename striatum y2
. rename temporalcortex y3
. rename thalamus y4
. rename posteriorcingulatecortex y5
. rename parietalcortex y6
. rename occipitalcortex y7
reshape long y, i(participant) j(region)
encode participant, gen(id)
generate logy = log(y)
# Simple regression
regress y fmrp
## Regression with adjustment for regions
regress y fmrp i.region, vce(ols)
# Adjusted plot
twoway (lfitci y fmrp) (scatter y fmrp), ytitle(mGluR_5_ Activity) xtitle(FMRP (ng/ul)) title(Adjusted Linear Regression of FMRP on mGluR_5_ Activity) legend(rows(3) position(4) ring(0))
# Simple plot
twoway (scatter y fmrp) (lfit y fmrp), ytitle(“mGluR_5_ Activity” “ ”) xtitle(FMRP (ng/ul)) title(Linear Regression of FMRP on mGluR_5_ Activity) legend(on order(1 “FM mGluR_5_ Activity” 2 “FXS-M mGluR_5_ Activity” 3 “Fitted Values” 4 “ ”) colfirst rows(3) size(vsmall) nobox lalign(outside) position(4) ring(0)) clegend(width(2) height(2))
## Making a table for publication
regress y fmrp
outreg2 using results, word replace ctitle(mGluR_5_ Activity, Without Adjustment) label dec(2) title(FMRP and mGluR_5_ Activity Regression) addnote()
regress y fmrp i.region, vce(ols)
outreg2 using results, word append ctitle(mGluR_5_ Activity, With Adjustment) label dec(2)

## Data Availability

The data presented in this study are openly available in [62]. Available online: https://doi.org/10.5281/zenodo.5792324 (accessed on 22 December 2021) [62].

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
