# Peer review of "Fragile X Mental Retardation Protein and Cerebral Expression of Metabotropic Glutamate Receptor Subtype 5 in Men with Fragile X Syndrome: A Pilot Study"

_brainsci, 2022, doi:10.3390/brainsci12030314_

Round 1

Reviewer 1 Report

Imaging biomarkers are a potentially very useful tool to assess disease severity and drug efficacy in fragile X syndrome. This paper uses a radiolabeled probe to assess mGluR5 levels in brain regions in humans with the full fragile X mutation by PET. The work is an extension of the authors’ published work in 2020 in Brain Sciences testing this probe in fragile X compared to typically developing individuals. There are a couple of issues that should be addressed: (1) Figure 2 has been previously published. (2) Although age and diurnal differences are discussed, there is no discussion of potential differences in FMRP expression as a function of brain region. This is important because FMRP levels here were determined by ELISA in lymphocytes, and peripheral and brain region expression may differ. Overall, the work has important implications for the mGluR Theory of Fragile X and future studies could positively impact clinical trials.

Author Response

We thank you for your suggestions. We have indicated our changes in italics. 

Point 1: Imaging biomarkers are a potentially very useful tool to assess disease severity and drug efficacy in fragile X syndrome. This paper uses a radiolabeled probe to assess mGluR5 levels in brain regions in humans with the full fragile X mutation by PET. The work is an extension of the authors’ published work in 2020 in Brain Sciences testing this probe in fragile X compared to typically developing individuals. There are a couple of issues that should be addressed: (1) Figure 2 has been previously published.

Response 1: We have included Figure 2 with permission from the copyright holders to reproduce it. Readers may be unfamiliar with the radiotracer so the figure provides visual evidence of density and distribution of the radiotracer in the brains of healthy men with typical development.

Point 2:(2) Although age and diurnal differences are discussed, there is no discussion of potential differences in FMRP expression as a function of brain region. This is important because FMRP levels here were determined by ELISA in lymphocytes, and peripheral and brain region expression may differ. Overall, the work has important implications for the mGluR Theory of Fragile X and future studies could positively impact clinical trials.

Response 2: We added a statement as the second to last paragraph under 4.1 Limitations as follows:

Another limitation is our use of ELISA in lymphocytes to estimate FMRP expression as a function of brain region. Peripheral FMRP expression may differ from FMRP expression in various brain regions. 

Reviewer 2 Report

The manuscript describes an association between FMRP levels in full mutated individuals and  mGluR5 expression in the anterior cingulate cortex, the occipital cortex, and the thalamus. This manuscript is based on previous work published in a previous paper by the same group in which they showed that mGluR5 expression was significantly reduced in cortical and subcortical regions of men with FXS in contrast to age-matched men with TD (Brain Sci. 202010(12), 899). 

I have had great difficulty following the text as it is very schematic and tables and figures are not self-explanatory.  Discussion is almost totally absent and also schematic and with very little content. The manuscript should be extended to make it more "reader-friendly", explaining the meaning of each table and figure and discussing in greater length their results and the importance of their findings in the context of what it means for treatment of patients with  mGluR-based therapies as this is not clear from their results. The article should be extensively rewritten.
.

Methodologically, the comparison of mGluR5 expression in different areas of the brain to FMRP levels in blood based on 6 individuals with extremely low levels of FMRP (as would be expected).  Based on this the authors conclude that there is a significant association of FMRP levels and mGluR5 expression.  The use of Fragile X Full mutated individuals to determine an association between levels of FMRP and mGluR5 expression seems to be counterintuitive as protein is basically absent in these individuals.  Maybe the use of mosaic full mutation/premutation individuals and premutated individuals together with full mutated individuals could give a gradation of FMRP levels that would make results more significant.  

Author Response

We thank you for your suggestions. We have indicated our changes in italics.   Point 1: The manuscript describes an association between FMRP levels in full mutated individuals and  mGluR5 expression in the anterior cingulate cortex, the occipital cortex, and the thalamus. This manuscript is based on previous work published in a previous paper by the same group in which they showed that mGluR5 expression was significantly reduced in cortical and subcortical regions of men with FXS in contrast to age-matched men with TD (Brain Sci. 2020, 10(12), 899).   I have had great difficulty following the text as it is very schematic and tables and figures are not self-explanatory.  Discussion is almost totally absent and also schematic and with very little content. The manuscript should be extended to make it more "reader-friendly", explaining the meaning of each table and figure and discussing in greater length their results and the importance of their findings in the context of what it means for treatment of patients with  mGluR-based therapies as this is not clear from their results. The article should be extensively rewritten.  

Response 1: Thank you for identifying the flaws in the representation of our submission. We have revised the manuscript as suggested to explain the revised tables and figures. We have expanded the discussion to point out the ways in which our findings provide the foundations for novel approaches for the diagnosis and treatment of people with FXS and related conditions. We have extensively rewritten the article accordingly. 

Point 2: Methodologically, the comparison of mGluR5 expression in different areas of the brain to FMRP levels in blood based on 6 individuals with extremely low levels of FMRP (as would be expected).  Based on this the authors conclude that there is a significant association of FMRP levels and mGluR5 expression.  The use of Fragile X Full mutated individuals to determine an association between levels of FMRP and mGluR5 expression seems to be counterintuitive as protein is basically absent in these individuals.  Maybe the use of mosaic full mutation/premutation individuals and premutated individuals together with full mutated individuals could give a gradation of FMRP levels that would make results more significant.   

Response 2: Thank you for providing insights to make sense of our data. We have included data from a person with an allele size mosaicism and CGG repeats approaching those of men with a full mutation to yield significant findings. The abstract, methods, results, and discussion have been extensively rewritten accordingly.

Round 2

Reviewer 1 Report

There are some typos, for example, in section 2.2.1 "FRMP". 

It would be helpful if colors with more contrast were used for Figure 3 (the dark aqua and black look the same).

Author Response

We thank you for your suggestions. We have indicated our changes in italics. 

Point 1: There are some typos, for example, in section 2.2.1 "FRMP". 

Response 1: We have corrected the typos.

Point 2: It would be helpful if colors with more contrast were used for Figure 3 (the dark aqua and black look the same).

Response 2: We have utilized colors with more contrast for Figure 3 as follows:

This manuscript is a resubmission of an earlier submission. The following is a list of the peer review reports and author responses from that submission.

Round 1

Reviewer 1 Report

Accepted

Reviewer 2 Report

The paper by Brašic and co-workers investigates the expression of mGLUR5 in the brain of FXS patients by using [18F]FPEB, a selective and safe radioligand of mGLUR5; in addition, it aims at correlate FMRP and mGLUR5 expression levels. Although, the paper is potentially interesting, it presents many shortcomings that make it not yet acceptable for the publication. The major problem is that results are inconclusive with respect to finding a correlation between FMRP expression and mGLUR5 reduction because of the small number of patients involved in the study; authors state that they want to perform further studies with an adequate sample size to solve this issue: considering that the results obtained about the reduction of mGLUR5 in the brain of FXS have been already published by Mody and co-workers, who used the same method here described, the novelty of the present study is the above mentioned correlation that, however, was not demonstrated; for this reason, authors should publish conclusive results after increasing number of patients involved.

In addition, other weaknesses can be revealed in the paper:

  • the authors state that finding a correlation between FMRP and mGLUR5 expressions can “provide a tool to apply precision medicine“, but they should better explain in which way this can be achieved.
  • Details about study subjects are lacking: were FXS patients receiving medications? If so, which type? Had all patients an FMRP full-mutation?
  • Details about processing of data obtained from PET should be shown.
  • A description of the method used to measure FMRP expression in FXS patients is completely lacking.
  • In Figure 2, Oc, Pa, pCg and Tp should be indicated also on the graph. The same for CN, Pu and Th in figure 3. Idem on figure 4.
  • p=0.16 on figure 6 cannot be considered a trend towards significance.

Reviewer 3 Report

The manuscript entitled “Fragile X mental retardation protein and cerebral expression of metabotropic glutamate receptor subtypes 5 in men with fragile X syndrome” deals with the important issue of clinical markers of neurodevelopmental disorders. Authors analysed the expression of mGluR5 receptors using [18F]FPEB radioligand in healthy controls and patients with FXS. Authors compared the expression of mGluR5 with the levels of FMRP. The main conclusion is that mGluR5 expression is reduced in the patients with FXS. After reading the manuscript I have several major concerns about the study design and presented results.

  1. The information about the tested subjects is very limited. Beside the age of participants nothing else is provided. Many other factors could influence the results (such as medication used). Inclusion and exclusion criteria are lacking and described very generally (eg. which abnormal laboratory values or significant unstable serious medical, neurological or psychiatric illnesses). Furthermore, there is no data about subjects’ cognitive status. As at least two FXS subjects (of 7) have values of mGluR5 within the range of TD subjects in all regions it would be interesting to see if any other factor correlates with these results.
  2. Many relevant pieces of information are not provided in the manuscript, instead the citation to the one version of this manuscript in public repository is provided. I personally feel, although such repositories are important for fast communication of data, it is crucial to include all the necessary data in the manuscript so that readers can fully comprehend the presented data and conclusions based on it. The differences between two groups are very big. Beside disparity in the number of tested individuals, the age difference can significantly impact the results of the study. Authors should have match subjects form both groups based on the relevant data (at least age, sex, BMI, etc.) to eliminate some of confounding factors. As authors noted in the limitations part mGluR5 expression decreases with age thus this needs to be accounted for when analysing data. Furthermore, some of the subjects are younger than 25 years; as we consider the brain to be fully mature between the age of 25 and 30 it is possible that expression of mGluR5 in these subjects is still influenced by the developmental processes.
  3. There is no data about radioligand included in this manuscript. Although readers could find relevant data in other publications, basic information should be included so that readers can evaluate data without searching the literature (e.g. half-life of the radiolignad and other relevant pharmacological parameters of the radiolignad).
  4. As there is a difference in testing protocols and measuring parameters between two sites the authors should provide the analysis showing no major differences between analysed protocols (as they claim) so that readers could evaluate the differences for themselves. I personally feel that in this kind of study a differences in protocols will impact the results.
  5. Authors analysed the mGluR5 in several brain regions (caudate nucleus, occipital cortex, parietal cortex, posterior cingulate cortex, putamen, thalamus, temporal lobe), however, they did not provide description of how aforementioned regions were delineated and what was exactly included in these regions (e.g. entire caudate nucleus or just head of the nucleus, which part of the occipital cortex or entire cortex, what were the borders of OCC analysed, what kind of images were used to delineate structure, etc.). Furthermore, authors claim that reduced mGluR5 could be basis for cognitive impairment of FXS patients, however authors did not analyse regions important for cognitive function. Majority of “higher cognitive functions” are located in the prefrontal cortex and when considering learning and emotions hippocampus and amygdala are very important structure. Authors did not analyse any of these structures. They should either include them in the analysis or describe the reasons for not including them and then reinterpret conclusions.
  6. The description of statistical procedures are severely lacking as there is no described procedures, test or parameters used except “dot plots and regression were used”. The statistical analysis need to be described in detail.
  7. The results are poorly described with very limited information. In fact the exactly the same formulations are used three times with change of one word (for analysed region). This is completely uninformative and useless. The statements as “There appears to be a positive correlation between mGluR5 uptake and (insert region) and FMRP concentrations in the cohorts of men with FXS with and without the cohorts of participants with TD” are unclear and more suited to discussion. In results data should be provided (in this case some numbers).
  8. Provided figures are not very informative. They provide the same data as tables without any additional information. It would be nice if data presented in figures were organized by the subjects or regions or both to get more clarity. Figure legends are also “copy-paste” and do not provide any important pieces of information.
  9. One of the biggest weaknesses in the manuscript is the lack of FMRP measurements in TD subjects. Authors based the levels of FMRP in TDs on 10 healthy individuals from the literature. To me this is completely wrong as the basis of analysis was to correlate mGluR5 with the FMRP in the same individual. The average could be correct however it cannot be correlated as FRMP is average (one value repeated) and mGluR5 results represent one individual. Authors should have measured FMRP in TD subjects. This is invasive procedure, however these individuals already agreed to be irradiated so I believe that taking samples for FMRP measurements would not be “deal-breaker” for them. I think that data obtained from these analyses are unreliable and should be dropped. One could argue for the inclusion of data from other studies if presented data were obtain from meta-analysis of literature and not from one 10 person sample (too many unknown factors can influence this group). Furtermore, there is no description of how the FMRP was measured in FXS subjects (methods, samples, protocols, chemicals used, etc.)

There are many other minor issues with the manuscript which I did not mention as these major issues seriously detract from the quality and importance of the manuscript.